# TRUST – TRANSFORMER-DRIVEN U-NET FOR SPARSE TARGET RECOVERY

## ABSTRACT

Many inverse problems—from coded aperture optics to undersampled MRI—operate with unknown or poorly characterized sensing operators $\mathbf{A}$. Yet most sparse recovery methods assume $\mathbf{A}$ is precisely known, forcing costly calibration or restrictive acquisition protocols. We address the more realistic setting in which only limited number of observation–target pairs $(\mathbf{y}, \mathbf{x})$ are available, necessitating *joint operator learning* and *signal reconstruction*. The core challenge is cross-domain dispersion: local structures in the signal $\mathbf{x}$ are spread globally into measurements $\mathbf{y} = \mathbf{A}\mathbf{x}$, while CNN architectures rely on local receptive fields. We propose **TRUST**, a *hybrid model that uses multi-resolution attention to recover sparse support directly from measurements*. Theoretically, under the standard RIP conditions on $\mathbf{A}$, we show that attention maps computed on $\mathbf{y}$ approximate those computed on the true signal $\mathbf{x}$, with error bounded by the RIP constant. Architecturally, a Vision Transformer encoder estimates global sparse support from $\mathbf{y}$, and attention-guided skip connections steer a U-Net decoder to concentrate reconstruction capacity on support-consistent regions, coupling global contexts with local details. TRUST resolves the mismatch between measurement dispersion and the locality bias of CNN-only approaches. Across optical imaging, FastMRI, and ImageNet benchmarks, it consistently surpasses strong baselines – both objectively and subjectively – with marked reductions in hallucination artifacts. These results establish attention-guided support estimation as a principled and effective approach to high-quality reconstruction while jointly learning unknown sensing operators, enabling robust performance on inverse problems where conventional methods require the precise knowledge of forward models.

## 1 INTRODUCTION

The linear inverse problem is fundamental to modern signal processing, statistical modeling, and machine learning. The typical model here is $\mathbf{y} = \mathbf{A}\mathbf{x} + \mathbf{w}$, where we seek to recover an unknown signal $\mathbf{x} \in \mathbb{R}^n$ from a set of potentially noisy measurements $\mathbf{y} \in \mathbb{R}^m$ using the sensing matrix or the sensing operator $\mathbf{A} \in \mathbb{R}^{m \times n}$. This problem arises in a wide range of scientific and engineering applications, including magnetic resonance imaging (MRI), computed tomography (CT), optical imaging, geophysics, astronomy and remote sensing, where observations are often limited, incomplete, noisy or partially corrupted (Tibshirani, 1996; Vogel, 2002a; Tarantola, 2005a; Ribes and Schmitt, 2008).

Classical approaches to solving inverse problems have been significantly advanced by the theory of compressed sensing (CS) and associated sparse recovery methods (Candès et al., 2006b; Donoho, 2006; Candès et al., 2006a; Elad, 2010). These techniques leverage the fact that many natural signals are sparse or compressible in specific transform domains, such as wavelets, gradients, or learned dictionaries. Under suitable conditions on the sensing matrix $\mathbf{A}$, CS guarantees accurate recovery of sparse signals from far fewer measurements than traditionally required. The reconstruction problem is typically posed as follows

$$\min_x \|\mathbf{x}\|_0 \quad \text{subject to} \quad \|\mathbf{A}\mathbf{x} - \mathbf{y}\|_2 \leq \epsilon \quad \text{or} \quad \min_x \|\mathbf{x}\|_1 \quad \text{subject to} \quad \|\mathbf{A}\mathbf{x} - \mathbf{y}\|_2 \leq \epsilon \quad (1)$$

where the $\ell_0$- or $\ell_1$-norm promotes sparsity in $\mathbf{x}$ and the constraint enforces fidelity to the measurements $\mathbf{y}$. While these methods are mathematically principled and offer performance guarantees,

they rely on accurate knowledge of the sensing operator $\mathbf{A}$ and assume linearity – assumptions that often break down in more complex or nonlinear measurement settings.

Deep learning has recently emerged as a powerful data-driven alternative to mitigate the limitations of classical approaches. In particular, convolutional neural networks (CNNs), notably encoder-decoder architectures like U-Net (Ronneberger et al., 2015a) have shown strong performance in tasks such as denoising (Zhang et al., 2017; 2018), super-resolution (Ledig et al., 2017) and compressive image recovery (Mousavi et al., 2015). These models learn to map raw sensor measurements directly to reconstructed signals, promising end-to-end inverse modeling, eliminating the need for hand-crafted priors, and enabling greater adaptability to real-world variations. This is particularly impactful in domains like synthetic aperture radar (SAR) and computational optics, where the forward process involves nonlinear physics such as diffraction or phase retrieval that are analytically intractable (Rivenson et al., 2018; Jin et al., 2017). These methods not only improve reconstruction quality, but also generalize well when trained on realistic measurement-target pairs.

Despite these advances, cross-domain inverse problems—where measurement and target domains are fundamentally different—remain a substantial challenge. For example, in optical systems, the relationship between observations and desired reconstructions is often nonlinear and ambiguous. Additionally, standard CNNs are inherently limited by their local receptive fields and spatial inductive biases, making it difficult to capture the global context and long-range dependencies essential for resolving such ambiguities. To overcome these limitations, researchers have begun exploring transformer-based architectures, which leverage self-attention mechanisms to model global interactions across spatial regions (Dosovitskiy et al., 2020; Chen et al., 2021). These models have shown remarkable success in high-level vision tasks and are increasingly being adopted in low-level inverse problems.

In this work, we introduce a novel architecture called TRUST, a transformer-driven U-Net for sparse target recovery that integrates the Vision Transformer (ViT) with U-Net for optical image reconstruction. Unlike only convolution blocks that primarily rely on local filtering, the attention mechanism successfully captures global dependencies across image patches, making them especially suited for cross-domain reconstruction tasks. Extensive experiments demonstrate that TRUST consistently outperforms traditional compressed sensing methods and state-of-the-art deep learning models.

## 2 PROBLEM DEFINITION

In this paper, we address the classical inverse problem $\mathbf{y} = A\mathbf{x} + \mathbf{w}$ via sparse recovery as in (1) under the challenging condition where *the sensing operator* $\mathbf{A}$ *is unknown* and *we only have access to a limited set of available observation-target pairs* $\{\mathbf{x}, \mathbf{y}\}$ *as training data*. Note that both the measured data $\mathbf{y}$ and the target images $\mathbf{x}$ are commonly flattened into vectors for mathematical convenience, although they originally represent structured two-dimensional spatial information.

Solving this ill-posed inverse problem using classical sparsity-driven methods would typically require first approximating the unknown operator $\mathbf{A}$ via dictionary learning techniques (Aharon et al., 2006), followed by applying sparse recovery algorithms such as Orthogonal Matching Pursuit (OMP) (Tropp and Gilbert, 2007) or the Fast Iterative Shrinkage-Thresholding Algorithm (FISTA) (Beck and Teboulle, 2009). However, this two-step approach is often inefficient, particularly in complex or nonlinear sensing environments (Tarantola, 2005b; Vogel, 2002b). As an alternative, we adopt modern deep learning-based strategies, specifically U-Net (Ronneberger et al., 2015a) and the proposed TRUST architecture, which directly learn the inverse mapping from data. These models eliminate the need for explicit knowledge of the sensing matrix while simultaneously enabling accurate reconstruction of sparse target signals (Mardani et al., 2019).

Throughout this paper, we motivate the development of the proposed TRUST network and illustrate its working concept in the context of a practical noninvasive coded aperture multicore fiber microendoscope for brain imaging (Willett et al., 2007; Farahi et al., 2013), capable of capturing sub-micron spatial image features.

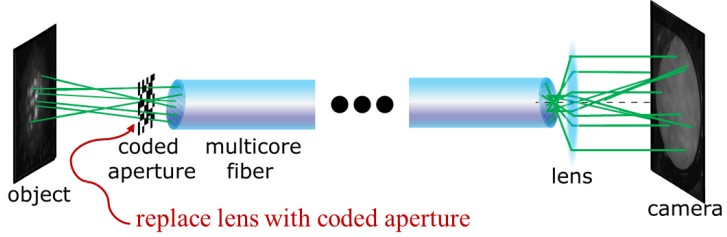

Figure 1: A multicore fiber coded aperture microendoscope. The fiber bundle contains around 6000 cores, has a diameter of 270 $\mu m$, capable of capturing sub-micron image features.

## 3 TRUST

### 3.1 RELATED WORKS

Numerous efforts have been made to address the sparse recovery problem using deep learning. Early pioneering approaches, such as ISTA-Net (Zhang and Ghanem, 2018) and ADMM-Net (Sun et al., 2016), belong to the class of algorithm unrolling methods (Monga et al., 2021). These architectures translate each iteration of a classical sparse optimization algorithm into a corresponding layer of a neural network, allowing the model to learn key parameters while preserving the interpretability of the original iterative structure. Although unrolling networks offer advantages in terms of interpretability, parameter efficiency, and performance in structured or low-data regimes, they generally fall short when applied to large-scale complex recovery tasks.

In contrast, more general-purpose architectures like U-Net have emerged as dominant solutions in signal and image reconstruction. Originally designed for biomedical image segmentation, U-Net's encoder–decoder structure with skip connections allows it to effectively capture and integrate multiscale features, making it well-suited for complex spatial reconstruction tasks (Ronneberger et al., 2015b). Recent advancements such as TransUNet (Chen et al., 2021) further enhance U-Net's capabilities by incorporating attention mechanisms at the network bottleneck, leveraging the strength of self-attention to model long-range dependencies and improve global context modeling. In the opposite direction is the fully transformer-based encoder–decoder Restormer (Zamir et al., 2022), which integrates attention mechanisms with multiscale architectures for image reconstruction.

A closer examination of the linear inverse problem $\mathbf{y} = \mathbf{A}\mathbf{x}$ reveals a fundamental challenge: *local features in the signal* $\mathbf{x}$ *may become dispersed or diffused across the global observation* $\mathbf{y}$. This is particularly true in compressed sensing, where measurements are often acquired in incoherent or randomized domains to satisfy theoretical recovery guarantees. In such settings, reconstruction architectures that primarily rely on local receptive fields—such as classical CNNs or even U-Net—can struggle to recover globally consistent structure, especially when long-range dependencies are critical to disambiguate spatial information.

### 3.2 PROPOSED ARCHITECTURE

Motivated by these limitations, we propose TRUST, a hybrid architecture designed to combine the strengths of both local and global modeling paradigms. As illustrated in Figure 2, TRUST employs a Vision Transformer (ViT) to extract multiscale global attention features from the input, effectively modeling long-range dependencies across the spatial domain. These features are then processed through an adaptive pooling layer, which performs pixel-wise smoothing to enhance robustness and feature continuity. Finally, a U-Net-inspired upsampling pathway incrementally refines the output, progressively recovering fine spatial detail and enforcing structural coherence.

In the remainder of this section, we delve into the design rationale behind each component of the TRUST architecture. We aim to provide a deeper understanding of their individual contributions and their synergistic effect on the network's overall performance in sparse recovery tasks.

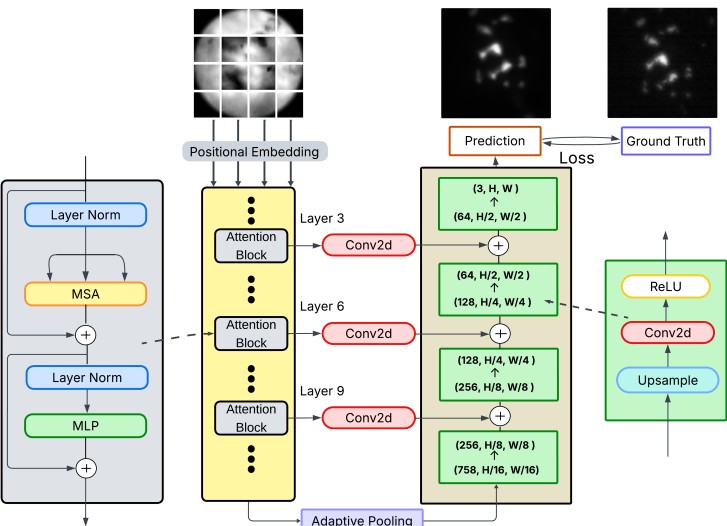

Figure 2: TRUST Architecture – Transformer-Driven U-Net for Sparse Target Recovery

### 3.3 ATTENTION CAN BE AN EXCELLENT ENCODER

Compared to traditional convolutional operations, the attention mechanism in Transformers offers a significant advantage in modeling global contextual relationships across spatial features. At the heart of this mechanism is the self-attention operation, defined as:

$$\text{Attention}(\mathbf{Q}, \mathbf{K}, \mathbf{V}) = \text{softmax}\left(\frac{\mathbf{Q}\mathbf{K}^T}{\sqrt{d_k}}\right)\mathbf{V} \tag{2}$$

where $\mathbf{Q}$, $\mathbf{K}$, and $\mathbf{V}$ denote the query, key, and value matrices, respectively, and $d_k$ is the dimensionality of the key vectors. This formulation effectively performs a scaled dot-product similarity – akin to a normalized cosine similarity – which allows the model to dynamically focus on salient regions and capture long-range structural dependencies across the entire image.

We further demonstrate that self-attention applied directly to the measurement domain $\mathbf{y}$ can approximate the attention features of the ground truth signal $\mathbf{x}$, provided that the sensing matrix satisfies the Restricted Isometry Property (RIP) (Candès and Tao, 2005). Specifically, if $\mathbf{A}$ satisfies the Restricted Isometry Property (RIP) of order $2k$ with RIP constant $\delta_{2k} \in (0, 1)$, then for all $2k$-sparse vectors $\mathbf{z} \in \mathbb{R}^n$, we have

$$(1 - \delta_{2k})\, \|\mathbf{z}\|_2^2 \ \le \ \|\mathbf{A}\mathbf{z}\|_2^2 \ \le \ (1 + \delta_{2k})\, \|\mathbf{z}\|_2^2.$$

This implies that the geometry of sparse vectors is approximately preserved under the mapping $\mathbf{A}$. More precisely, the attention error between two representations in two different domains is bounded by the RIP constant as follows (see the Appendix for the detailed derivation):

$$\left|\mathbf{y}^\top \mathbf{y}' - \mathbf{x}^\top \mathbf{x}'\right| = \left|\mathbf{x}^\top \mathbf{A}^\top \mathbf{A}\mathbf{x}' - \mathbf{x}^\top \mathbf{x}'\right| \ \le \ \delta_{2k}.$$

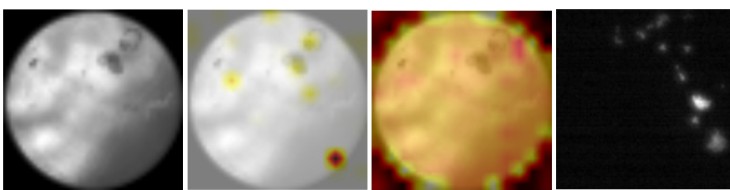

Figure 3: Overlaying attention map of a sample collected from the microendoscope in Figure 1. From left to right: response $\mathbf{y}$, single head attention, aggregated multihead attention, and ground truth $\mathbf{x}$.

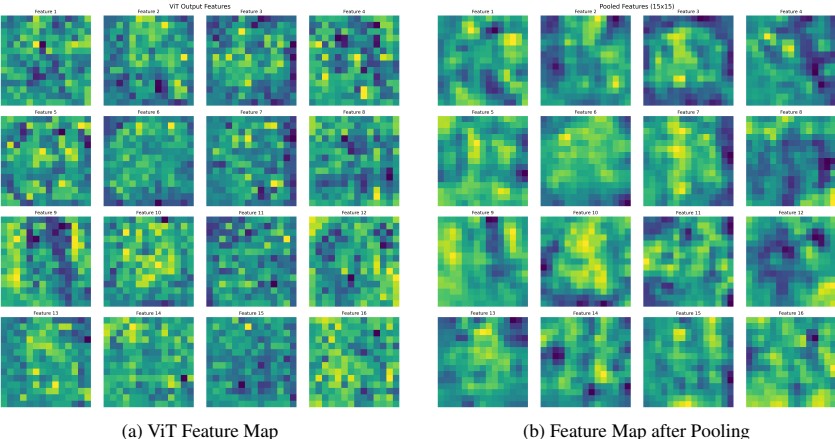

(a) ViT Feature Map                    (b) Feature Map after Pooling

Figure 4: Adaptive pooling layer function's effect on a typical attention map.

As depicted in Figure 3, the attention map generated from $\mathbf{y}$ indeed highlights key spatial structures and regions that closely resemble those in the original image $\mathbf{x}$. This empirical observation aligns with our theoretical analysis and confirms that the attention module not only facilitates contextual reasoning, but also plays a critical role in sparse support recovery. These extracted attention features serve as a powerful prior, guiding the subsequent reconstruction stages in our TRUST framework to focus on the most informative regions of the measurement.

### 3.4 ADAPTIVE POOLING LAYER

Processing full-resolution attention maps is costly and misaligned with spatial hierarchies. We therefore insert an *adaptive pooling* layer for (i) dimensionality reduction and (ii) feature standardization: it compresses the attention output to a coarser, semantically focused resolution and normalizes it to a fixed size regardless of input shape (He et al., 2015). As shown in Figure 4, this distillation preserves structure while yielding a compact representation, enabling more efficient and precise decoding.

### 3.5 U-NET-LIKE UPSAMPLING DECODER FOR DETAIL REFINEMENT

The decoder reconstructs high-resolution images from the pooled feature maps using a U-Net–style design: each stage upsamples to restore spatial resolution, then applies Conv2D layers with ReLU to refine structure and add nonlinearity. This stage-wise refinement progressively recovers fine details that were compressed or diffused during encoding.

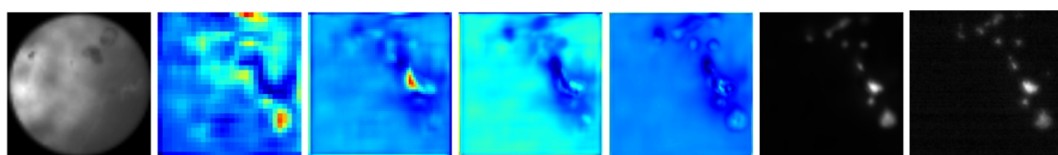

Figure 5: Different stages of decoding. From left to right: response $\mathbf{y}$, stage 1, stage 2, stage 3, stage 4, reconstructed image $\hat{\mathbf{x}}$, and ground truth $\mathbf{x}$. Resolution is enhanced gradually from left to right.

As shown in Figure 5, we track feature maps through the decoder. The raw diffraction pattern is transformed by attention and convolutions to reveal structure. At the first decoding stage, high-frequency components emerge (strong activations in red/yellow). Subsequent layers increase spatial resolution while reducing channels, reconstructing the signal's hierarchy.

This visualization shows how the model bridges incoherent measurements and target images: the Transformer captures global dependencies early, and the U-Net decoder restores local structure via multiscale upsampling. Activation evolution indicates selective amplification of salient features

Table 1: Unified results with task in the leftmost column. Metrics are mean $\pm$ std. Higher is better for PSNR/SSIM; lower is better for MAE/MSE/FPR/Time. Best performance is in **red**. Per $16 \times 16$ patch, we apply a fixed randomly Gaussian orthonormal transform and random keep 25% or 100% pixels.

| Dataset | Model | PSNR (dB)↑ | SSIM↑ | MAE↓ | MSE↓ | FDR ($\times 10^{-2}$)↓ | Recon. Time (ms)↓ |
|---|---|---|---|---|---|---|---|
| ImageNet (100% Preserved + Mask) | TRANSUNET | 21.75 ± 2.89 | 0.539 ± 0.142 | 0.064 ± 0.025 | 0.008 ± 0.006 | 0.11 ± 0.2 | 9.7 ± 3.1 |
| | UNET | 27.19 ± 4.01 | 0.922 ± 0.039 | 0.039 ± 0.025 | 0.003 ± 0.005 | 0.02 ± 0.000 | 4.3 ± 2.1 |
| | RESTORMER | 28.27 ± 4.33 | 0.934 ± 0.028 | 0.036 ± 0.025 | 0.003 ± 0.004 | 0.007 ± 0.000 | 52.3 ± 4.1 |
| | TRUST | 28.27 ± 4.33 | 0.934 ± 0.028 | 0.036 ± 0.025 | 0.003 ± 0.004 | 0.002 ± 0.000 | 4.4 ± 2.3 |
| ImageNet (25% Preserved + Mask) | TRANSUNET | 7.35 ± 1.93 | 0.120 ± 0.055 | 0.374 ± 0.097 | 0.202 ± 0.088 | 2.23 ± 2.6 | 9.5 ± 4.1 |
| | UNET | 8.34 ± 2.05 | 0.174 ± 0.071 | 0.327 ± 0.088 | 0.163 ± 0.078 | 6.30 ± 6.61 | 4.4 ± 3.2 |
| | RESTORMER | 13.52 ± 2.14 | 0.378 ± 0.134 | 0.166 ± 0.043 | 0.050 ± 0.025 | 3.9 ± 4.4 | 50.6 ± 3.9 |
| | TRUST | 16.59 ± 1.94 | 0.347 ± 0.067 | 0.166 ± 0.096 | 0.042 ± 0.085 | 1.3 ± 1.9 | 4.5 ± 2.0 |
| FastMRI Reconstruction | OMP | 14.37 ± 4.34 | 0.145 ± 0.0395 | 0.138 ± 0.0923 | 0.109 ± 0.543 | 6.26 ± 3.22 | ~12,000 |
| | U-Net | 21.70 ± 2.74 | 0.668 ± 0.0900 | 0.0506 ± 0.0174 | 0.0861 ± 0.0246 | 4.26 ± 4.99 | 6.3 ± 2.2 |
| | TransUNet | 21.07 ± 2.34 | 0.6553 ± 0.0863 | 0.0396 ± 0.0178 | 0.0703 ± 0.0208 | 5.93 ± 6.21 | 13.2 ± 4.2 |
| | Restormer | 23.72 ± 3.15 | 0.698 ± 0.0953 | 0.0411 ± 0.0160 | 0.0692 ± 0.0227 | 2.97 ± 4.74 | 63.4 ± 8.3 |
| | TRUST | 24.81 ± 3.13 | 0.717 ± 0.0851 | 0.0353 ± 0.0133 | 0.0613 ± 0.0220 | 2.78 ± 4.33 | 11.2 ± 3.1 |
| Optics Reconstruction | OMP | 68.04 ± 2.03 | 0.279 ± 0.035 | 0.0435 ± 0.0062 | 0.0111 ± 0.0032 | 5.30 ± 1.03 | ~15,000 |
| | U-Net | 70.76 ± 2.00 | 0.772 ± 0.053 | 0.0398 ± 0.012 | 0.00451 ± 0.0022 | 1.14 ± 0.16 | 7.1 ± 2.0 |
| | TransUNet | 69.84 ± 1.92 | 0.636 ± 0.091 | 0.0440 ± 0.012 | 0.00911 ± 0.0040 | 2.61 ± 3.10 | 15.2 ± 3.9 |
| | Restormer | 70.48 ± 2.13 | 0.715 ± 0.056 | 0.0405 ± 0.013 | 0.00823 ± 0.0041 | 0.907 ± 0.36 | 68.4 ± 7.3 |
| | TRUST | 71.992 ± 1.94 | 0.814 ± 0.069 | 0.0253 ± 0.0073 | 0.00431 ± 0.0013 | 0.901 ± 0.22 | 12.2 ± 3.7 |

and suppression of noise, yielding high-fidelity reconstructions—combining global context with localized detail critical for robust sparse inverse recovery.

## 4 EXPERIMENTS

We leverage transfer learning on our proposed TRUST architecture by incorporating the pretrained 'google/vit-base-patch16-224' Vision Transformer as the encoder backbone (Dosovitskiy et al., 2020). This strategic choice significantly accelerates training convergence and improves performance for the specialized task of optical image reconstruction. Training was conducted on a setup with four Tesla P400 GPUs (24 GB VRAM each), using a learning rate of $1 \times 10^{-4}$ and a batch size of 128 and the inference is on NVIDIA 4070. Given the modest computational resources, training was extended over the course of one week to ensure stable convergence and optimal reconstruction quality.

### 4.1 DATASETS AND EVALUATION METRICS

We evaluate TRUST on three datasets—masked ImageNet, a custom optical set from the multicore fiber microendoscope (Figure 1), and single-coil knee FastMRI—covering both domain-specific reconstruction and standard inverse imaging. We report MSE, MAE, PSNR, SSIM, and False Discovery Rate (FDR) (Wang et al., 2004; Gonzalez and Woods, 2002); metric definitions and preprocessing/sampling-mask details are provided in the Appendix.

### 4.2 IMAGENET RESULTS

For ImageNet experiments, we curated a dogs-and-cats subset for training and validation and retained a disjoint holdout set for final testing. The training split contains 10,000 paired samples (orthogonally transformed patches and their ground-truth originals), with 1,000 pairs for validation and 1,000 for testing. All images were resized to $224 \times 224$ and partitioned into non-overlapping $16 \times 16$ patches; each patch was mapped by a fixed $256 \times 256$ orthonormal matrix, yielding a controlled inverse problem in which the model reconstructs the original image from its transformed representation. We then consider a harder setting: using the same fixed mask and randomly retains 25% of pixels per $16 \times 16$ patch (i.e., randomly drops 75%), effectively compressing each patch to $8 \times 8$ and producing a $112 \times 112$ masked measurement image. Reconstruction is performed from this masked domain back to the original resolution. Results for both settings are reported in Table 1 and Figure 6. Moreover, the trained model generalizes beyond the dogs-and-cats subset: it can reconstruct images from other semantic classes with the same forward operator and masking scheme, without any additional training as shown in Figure 13.

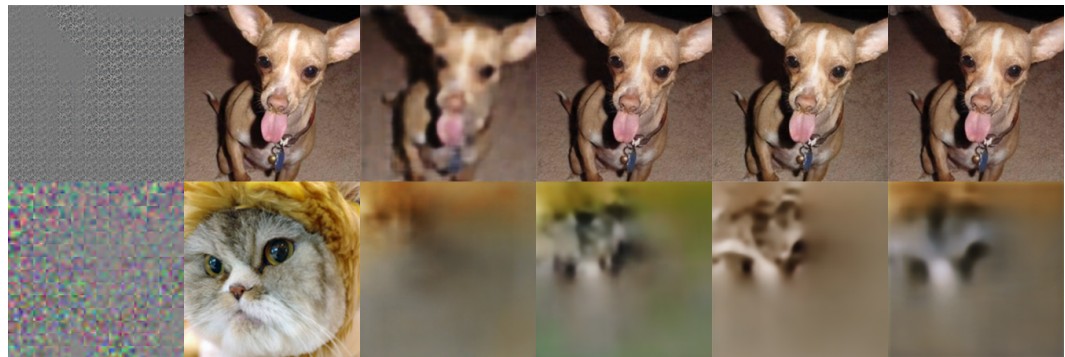

Figure 6: Different reconstruction results with corresponding SSIM and PSNR values. **Top Row** (left to right): 100% preserved maksed GT, GT, TransUnet $\{0.642, \ 18.638dB\}$, U-Net $\{0.682, \ 20.749dB\}$, Restormer $\{0.697, \ 20.187dB\}$ and TRUST $\{0.698, \ 21.786dB\}$. **Bottom Row** (left to right): 25% preserved maksed GT, GT, TransUnet $\{0.312, \ 11.437dB\}$, U-Net $\{0.361, \ 12.301dB\}$, Restormer $\{0.371, \ 12.283dB\}$ and TRUST $\{0.393, \ 13.021dB\}$.

## 4.3 MRI RESULTS

To demonstrate the generalization capability of TRUST, we conducted additional experiments on the FastMRI dataset – a large-scale benchmark jointly developed by Facebook AI Research and NYU Langone Health for accelerated MRI reconstruction (Zbontar et al., 2018). This task fits the ill-posed inverse problem described in Section 2, where the collected observation comes from an undersampled k-space signal processed through a sparse sampling operator $\mathbf{A}$. The degraded image, obtained via inverse Fourier transform (IFFT), contains aliasing artifacts. The goal is to reconstruct a high-quality ground truth image from this undersampled and noisy input (Lustig et al., 2007).We tested its performance on the large-scale standardized FastMRI dataset. Table 1 summarizes the results across 36 randomly selected slices from 108 subjects, totaling approximately 3,000 test images, whereas Figure 18 depicts a typical reconstruction sample.

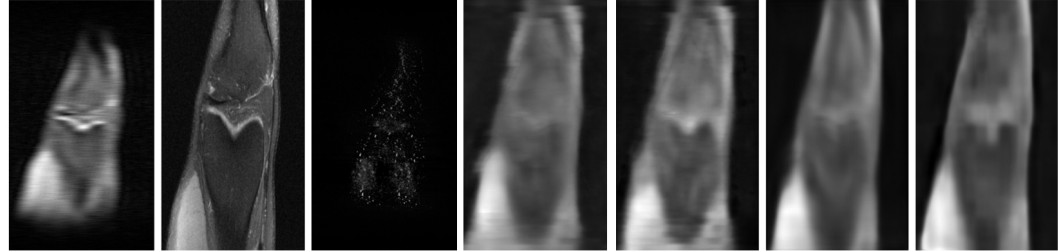

Figure 7: Example of reconstruction results with corresponding SSIM and PSNR values. From left to right: undersampled input $\mathbf{y}$, target $\mathbf{x}$, OMP $\{0.173, 15.682dB\}$, U-Net $\{0.610, 21.623dB\}$, TransUnet $\{0.614, 21.956dB\}$, Restormer $\{0.623, 22.631dB\}$, and TRUST $\{0.629, 22.893dB\}$
.

## 4.4 OPTICS RESULTS

For the optical dataset, training images came from two neuron slides and testing from a third, unseen slide: 10,000 response–target pairs for training and 5,000 for testing, all at an object-to-tip distance of $100\,\mu$m. This split tests generalization to new structures under matched conditions.

We compared TRUST with classical sparse recovery and deep learning baselines. U-Net and TRUST were trained with a joint $\ell_2$+SSIM loss using matched hyperparameters/budgets; sensitivity to the loss is discussed in Section 2.

From Table 1, TRUST surpasses U-Net and classical baselines on the 5,000-sample test set, yielding fewer hallucinations/artifacts. Visually, U-Net hallucinates structure near the bottom-left in a sample, while TRUST suppresses it and recovers a more faithful reconstruction.

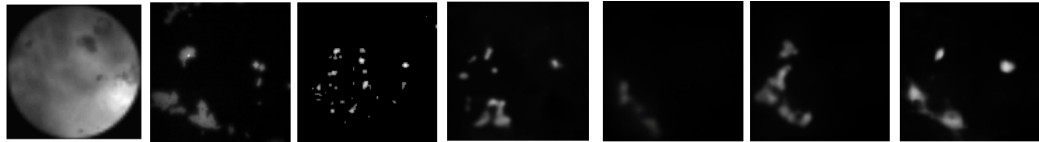

Figure 8: Example of reconstruction results with corresponding SSIM and PSNR values. From left to right: response $\mathbf{y}$, target $\mathbf{x}$, OMP $\{0.325, 63.071dB\}$, U-Net $\{0.636, 66.712dB\}$, TransUnet $\{0.553, 66.351dB\}$, Restormer $\{0.625, 66.583dB\}$, and TRUST $\{0.671, 68.276dB\}$
.

## 4.5 ABLATION STUDY

We study three factors affecting TRUST's reconstruction quality: (i) training loss, (ii) skip connections, and (iii) ViT pretraining. Unless noted, metrics are mean $\pm$ std over the test set.

**Loss function.** We compare $\ell_2$, $\ell_2+\ell_1$, and $\ell_2+$SSIM. While $\ell_2$ targets pixel fidelity and $\ell_1$ adds outlier robustness, SSIM optimizes structural similarity. As shown in Figure 9 and Table 2, $\ell_2+$SSIM yields the best overall MSE/MAE, PSNR, SSIM, and FDR, consistent with perceptual-loss findings (Zhao et al., 2016).



Figure 9: Reconstructions under different losses (SSIM, PSNR in dB). Left→right: $\mathbf{y}$, $\ell_2$ $\{0.137, 48.756\}$, $\ell_2+\ell_1$ $\{0.251, 67.693\}$, $\ell_2+$SSIM $\{0.798, 73.012\}$, and $\mathbf{x}$.

Table 2: Reconstruction performance under different training losses.

| Loss Function | MSE | MAE | PSNR (dB) | SSIM | FDR ($\times 10^{-2}$) |
|---|---|---|---|---|---|
| $\ell_2$ | $0.111 \pm 0.25$ | $0.318 \pm 0.073$ | $49.69 \pm 3.01$ | $0.101 \pm 0.0148$ | $1.057 \pm 0.64$ |
| $\ell_2 + \ell_1$ | $0.0101 \pm 0.18$ | $0.0797 \pm 0.092$ | $67.083 \pm 2.15$ | $0.243 \pm 0.053$ | $1.055 \pm 0.41$ |
| **$\ell_2$ + SSIM** | **$0.00431 \pm 0.0013$** | **$0.0253 \pm 0.0073$** | **$71.992 \pm 1.94$** | **$0.814 \pm 0.069$** | **$0.901 \pm 0.22$** |

**Skip connections.** To assess encoder–decoder shortcuts, we disable skips at various stages. Figure 10 and Table 3 show that removing even one skip degrades all metrics—most around edges/high frequencies—highlighting their importance during upsampling (Mao et al., 2016; He et al., 2016).

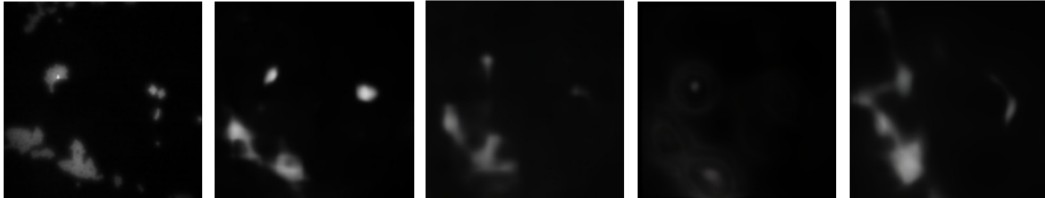

Figure 10: Removing skip connections (SSIM, PSNR in dB). Left→right: $\mathbf{x}$, TRUST $\{0.862, 72.744\}$, mv skip1 $\{0.610, 71.662\}$, mv skip1&2 $\{0.304, 67.832\}$, and no skip $\{0.654, 69.512\}$.

Table 3: Impact of skip connections.

| Configuration | MSE | MAE | PSNR (dB) | SSIM | FDR ($\times 10^{-2}$) |
|---|---|---|---|---|---|
| **TRUST** | **0.00431 ± 0.0013** | **0.0253 ± 0.0073** | **71.992 ± 1.94** | **0.814 ± 0.069** | **0.901 ± 0.22** |
| **TRUST mv skip1** | 0.00441 ± 0.0027 | 0.0280 ± 0.011 | 71.082 ± 1.91 | 0.774 ± 0.065 | 1.223 ± 0.28 |
| **TRUST mv skip1 & skip2** | 0.00681 ± 0.0046 | 0.0468 ± 0.023 | 70.156 ± 2.18 | 0.610 ± 0.1322 | 3.034 ± 0.64 |
| **TRUST no skip** | 0.00540 ± 0.0021 | 0.0314 ± 0.011 | 70.990 ± 1.80 | 0.746 ± 0.062 | 1.640 ± 0.47 |

**Pretraining.** We train the attention encoder from scratch vs. initializing from `google/vit-base-patch16-224`. Pretraining provides stronger, more general features and improves convergence and downstream accuracy on limited-domain data (Chen et al., 2019), with consistent gains across all metrics (Figure 11, Table 4).

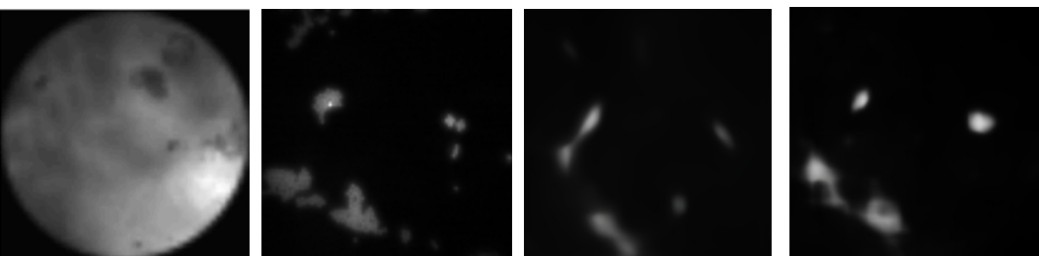

Figure 11: Pretraining vs. scratch (SSIM, PSNR in dB). Left→right: target, w/o pretraining $\{0.606, 71.342\}$, w/ pretraining $\{0.862, 72.744\}$.

Table 4: Effect of ViT pretraining.

| Method | MSE | MAE | PSNR (dB) | SSIM | FDR ($\times 10^{-2}$) |
|---|---|---|---|---|---|
| **TRUST without Pretrained ViT** | 0.00601 ± 0.0034 | 0.0341 ± 0.014 | 70.583 ± 1.81 | 0.697 ± 0.072 | 2.093 ± 0.19 |
| **TRUST with Pretrained ViT** | **0.00431 ± 0.0013** | **0.0253 ± 0.0073** | **71.992 ± 1.94** | **0.814 ± 0.069** | **0.901 ± 0.22** |

## 5 CONCLUSION AND FUTURE WORK

In this paper, we introduced TRUST, a hybrid architecture that integrates a pretrained Vision Transformer (ViT) encoder with a U-Net decoder for high-quality sparse image reconstruction. Experimental results show that TRUST consistently outperforms both classical and deep learning baselines, achieving superior performance across standard metrics, including PSNR, SSIM, MSE, MAE, and FDR, while significantly reducing hallucination artifacts.

TRUST's effectiveness is attributed to its key architectural components: *(i)* a ViT-based attention encoder that captures global dependencies early in the pipeline; *(ii)* skip connections that enable multi-scale feature fusion; and *(iii)* a hierarchical decoder that refines coarse global representations into high-resolution image details. Despite its advantages, TRUST introduces additional computational overhead due to its reliance on a pretrained transformer backbone, resulting in $2-3\times$ higher inference time compared to U-Net under equivalent hardware conditions. Also, while this study focuses on sparse optical image recovery, the underlying design principles of TRUST – attention-guided global context modeling and hierarchical multiresolution decoding – are broadly applicable (Touvron et al., 2021). Future work will explore TRUST extensions to various signal processing tasks while also addressing the model's computational complexity to improve efficiency and scalability (Mehta and Rastegari, 2022).

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

# APPENDIX

## A  ERROR BOUND FOR THE ATTENTION MECHANISM

We assume that we have two tokens $\mathbf{x}$ and $\mathbf{y}$, which are related via the linear constraint $\mathbf{y} = \mathbf{A}\mathbf{x}$. In practice, most of the time we have some additional prior knowledge on the operator $\mathbf{A}$ (after all, we typically design an appropriate $\mathbf{A}$ for the application at hand) such as:

- $\mathbf{A}$ is orthonormal square matrix; or

- $\mathbf{A}$ is tall matrix with orthonormal columns; or

- $\mathbf{A}$ is fat matrix satisfying the Restricted Isometry Property (RIP).

The attention mechanism is formulated as

$$\text{Attention}(\mathbf{Q}, \mathbf{K}, \mathbf{V}) = \text{softmax}\left(\frac{\mathbf{Q}\mathbf{K}^T}{\sqrt{d_k}}\right)\mathbf{V} \tag{3}$$

Performing self attention on $\mathbf{y}$ yields the following:

$$\text{Attention}(\mathbf{y}) = \text{softmax}\left(\frac{\mathbf{y}^T\mathbf{y}}{\sqrt{d_k}}\right)\mathbf{V} = \text{softmax}\left(\frac{\mathbf{x}^T\mathbf{A}^T\mathbf{A}\mathbf{x}}{\sqrt{d_k}}\right)\mathbf{V}. \tag{4}$$

When $\mathbf{A}$ has orthonormal columns, it is clear that attention above yields the same value in either $\mathbf{x}$ or $\mathbf{y}$ domain. In compressed sensing applications, $\mathbf{A}$ is most likely fat and the orthonormal property of its columns breaks down. In this case, we need to rely on the RIP of $\mathbf{A}$ as follows: let $\mathbf{A} \in \mathbb{R}^{m \times n}$ be a matrix satisfying the Restricted Isometry Property (RIP) of order $2k$ with constant $\delta_{2k} \in (0, 1)$. That is, for all $2k$-sparse vectors $\mathbf{z} \in \mathbb{R}^n$, we have

$$(1 - \delta_{2k})\|\mathbf{z}\|_2^2 \leq \|\mathbf{A}\mathbf{z}\|_2^2 \leq (1 + \delta_{2k})\|\mathbf{z}\|_2^2.$$

Let $\mathbf{x}, \mathbf{x}' \in \mathbb{R}^n$ be two normalized vectors with supports of size at most $k$, i.e., both are $k$-sparse and $\|\mathbf{x}\|_2^2 = \|\mathbf{x}'\|_2^2 = 1$. Then, their sum or difference support together has size at most $2k$. In other words, $\mathbf{x} + \mathbf{x}'$ and $\mathbf{x} - \mathbf{x}'$ are $2k$-sparse. We aim to bound the following difference between the original and transformed inner product:

$$\left|\mathbf{x}^\top\mathbf{A}^\top\mathbf{A}\mathbf{x}' - \mathbf{x}^\top\mathbf{x}'\right|.$$

The polarization identity combined with the RIP condition yields:

$$\|\mathbf{A}(\mathbf{x} + \mathbf{x}')\|_2^2 = \|\mathbf{A}\mathbf{x}\|_2^2 + 2\mathbf{x}^\top\mathbf{A}^\top\mathbf{A}\mathbf{x}' + \|\mathbf{A}\mathbf{x}'\|_2^2,$$
$$\|\mathbf{A}(\mathbf{x} - \mathbf{x}')\|_2^2 = \|\mathbf{A}\mathbf{x}\|_2^2 - 2\mathbf{x}^\top A^\top\mathbf{A}\mathbf{x}' + \|\mathbf{A}\mathbf{x}'\|_2^2.$$

Subtracting these two identities gives:

$$\|\mathbf{A}(\mathbf{x} + \mathbf{x}')\|_2^2 - \|\mathbf{A}(\mathbf{x} - \mathbf{x}')\|_2^2 = 4\mathbf{x}^\top\mathbf{A}^\top\mathbf{A}\mathbf{x}'.$$

Similarly, if $\mathbf{A}$ is the identity matrix, we have:

$$\|\mathbf{x} + \mathbf{x}'\|_2^2 - \|\mathbf{x} - \mathbf{x}'\|_2^2 = 4\mathbf{x}^\top\mathbf{x}'.$$

Imposing RIP on $\mathbf{x} + \mathbf{x}'$ and $\mathbf{x} - \mathbf{x}'$ produces

$$\left|\|\mathbf{A}(\mathbf{x} + \mathbf{x}')\|_2^2 - \|\mathbf{x} + \mathbf{x}'\|_2^2\right| \leq \delta_{2k}\|\mathbf{x} + \mathbf{x}'\|_2^2,$$

$$\left|\|\mathbf{A}(\mathbf{x} - \mathbf{x}')\|_2^2 - \|\mathbf{x} - \mathbf{x}'\|_2^2\right| \leq \delta_{2k}\|\mathbf{x} - \mathbf{x}'\|_2^2.$$

Combining the two and applying the triangle inequality, we can finally obtain the following bound:

$$
\begin{aligned}
\left|\mathbf{x}^\top \mathbf{A}^\top \mathbf{A} \mathbf{x}' - \mathbf{x}^\top \mathbf{x}'\right| &= \frac{1}{4} \left|\left(\|\mathbf{A}(\mathbf{x}+\mathbf{x}')\|_2^2 - \|\mathbf{A}(\mathbf{x}-\mathbf{x}')\|_2^2\right) - \left(\|\mathbf{x}+\mathbf{x}'\|_2^2 - \|\mathbf{x}-\mathbf{x}'\|_2^2\right)\right| \\
&\leq \frac{1}{4} \left(\left|\|\mathbf{A}(\mathbf{x}+\mathbf{x}')\|_2^2 - \|\mathbf{x}+\mathbf{x}'\|_2^2\right| + \left|\|\mathbf{A}(\mathbf{x}-\mathbf{x}')\|_2^2 - \|\mathbf{x}-\mathbf{x}'\|_2^2\right|\right) \\
&\leq \frac{\delta_{2k}}{4} \left(\|\mathbf{x}+\mathbf{x}'\|_2^2 + \|\mathbf{x}-\mathbf{x}'\|_2^2\right) \\
&= \frac{\delta_{2k}}{4} \left(2\|\mathbf{x}\|_2^2 + 2\|\mathbf{x}'\|_2^2\right) \\
&= \frac{\delta_{2k}}{2} \left(\|\mathbf{x}\|_2^2 + \|\mathbf{x}'\|_2^2\right) \\
&= \frac{\delta_{2k}}{2} \left(1 + 1\right) \\
&= \delta_{2k}.
\end{aligned}
$$

Figure 12 illustrates the average effect of sparsity and fat random Gaussian matrices on attention/similarity averaged over 100 totally random trials. As expected, $\mathbf{A}$'s with orthonormal columns yield exactly the same attention. On the other hand, we confirm that we are still able to obtain close approximation of the attention level with fat random Gaussian sensing matrices $\mathbf{A}$'s.

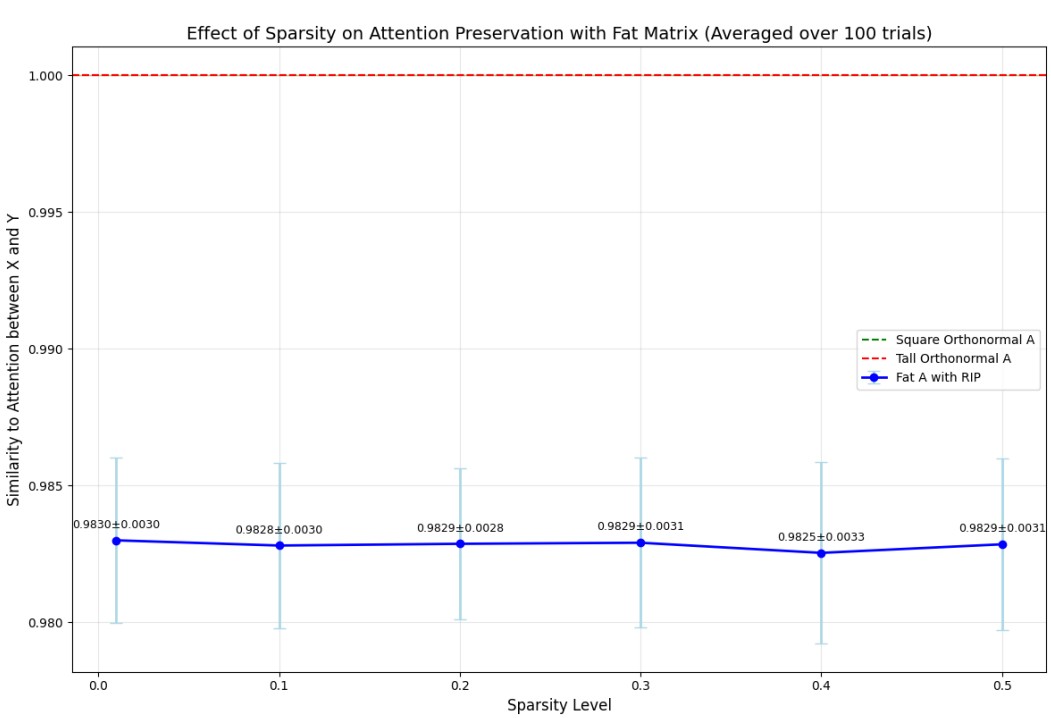

Figure 12: Simulation of similarity between attention on $\mathbf{x}$ and $\mathbf{y} = \mathbf{A}\mathbf{x}$ for various sensing matrices $\mathbf{A}$'s.

## B EVALUATION METRICS

To evaluate the reconstruction quality of our models, we employ both standard image similarity metrics and a custom hallucination-aware metric:

**Root Mean Squared Error (RMSE).** RMSE measures the square root of the average squared differences between predicted and ground truth pixel values:

$$\text{RMSE} = \sqrt{\frac{1}{N}\sum_{i=1}^{N}(x_i - \hat{x}_i)^2},$$

where $x_i$ and $\hat{x}_i$ are the ground truth and predicted pixel values, respectively.

**Peak Signal-to-Noise Ratio (PSNR).** PSNR quantifies the reconstruction fidelity relative to the maximum pixel intensity:

$$\text{PSNR} = 20 \cdot \log_{10}\left(\frac{\text{MAX}}{\text{RMSE}}\right),$$

where MAX is the maximum possible pixel value (assumed to be 1.0 after normalization).

**Structural Similarity Index Measure (SSIM).** SSIM evaluates perceptual image similarity by comparing local patterns of luminance, contrast, and structure. The score ranges from $-1$ to 1, with 1 indicating perfect structural alignment.

**False Positive Region Score (FPR).** We define a hallucination-sensitive metric called the False Positive Region (FPR) score to quantify spurious regions generated by the model. A pixel is considered hallucinated if it satisfies:

$$x_{\text{hat}} > t_{\text{high}} \quad \text{and} \quad x_{\text{true}} \le t_{\text{low}},$$

The FPR score is computed as the fraction of hallucinated pixels over the entire image:

$$\text{FPR} = \frac{\left|\{i : x_{\text{hat},i} > t_{\text{high}} \ \wedge \ x_{\text{true},i} \le t_{\text{low}}\}\right|}{N}.$$

## C EXTENDED SPARSE RECOVERY RESULTS

All the models listed below were trained with approximately same hyper-parameters as specified in the paper, and the stop condition is when reaching the nearly same loss values. This setup ensures a fair comparison under similar consistent conditions.

### C.1 EXTENDED RESULTS ON IMAGENET

We found that even the model is trained on cat/dogs dataset, still it can recover other category images.

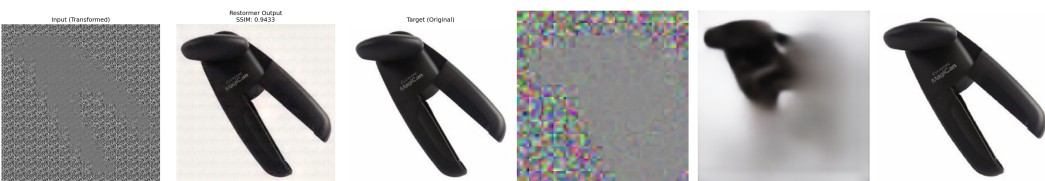

Figure 13: Other category reconstruction example

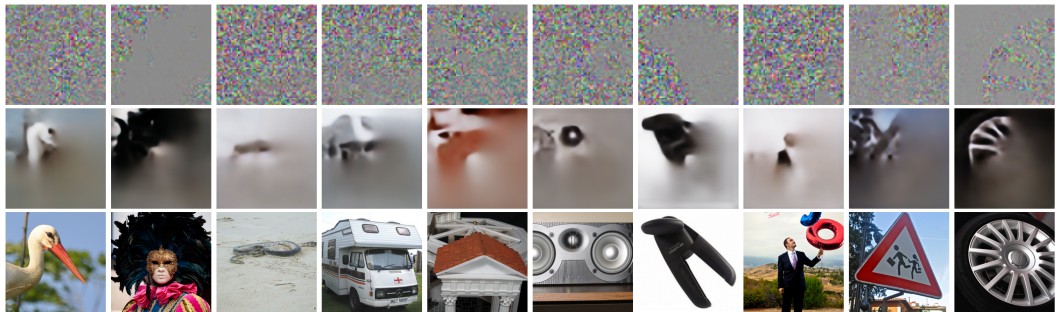

Figure 14: Large gird reconstruction by TRUST

## C.2 EXTENDED RESULTS ON SPARSE RECOVERY OF OPTICS DATA

In this section, we present a more comprehensive comparison of model performance on sparse recovery tasks using the optical imaging dataset.

Figures 15, 16, and 17 illustrate qualitative reconstruction results across various models, while the quantitative metrics are summarized in Table 5. The data clearly show that TRUST consistently outperforms all competing neural network architectures, achieving superior reconstruction fidelity across all evaluation criteria.

As expected, traditional sparse recovery methods deliver the weakest performance, producing reconstructions with significant artifacts and loss of structural detail. Among deep learning models, the fully transformer-based Restormer yields competitive results but exhibits a consistent tendency to under-predict fine-scale features, leading to a higher missing probability error. This suggests that despite its strong global modeling capabilities, Restormer may struggle to capture the fine-grained spatial details necessary for precise optical reconstruction.

These results reinforce the advantage of TRUST's hybrid architecture, which leverages both global attention mechanisms and localized multi-scale refinement to achieve accurate and perceptually faithful image recovery.

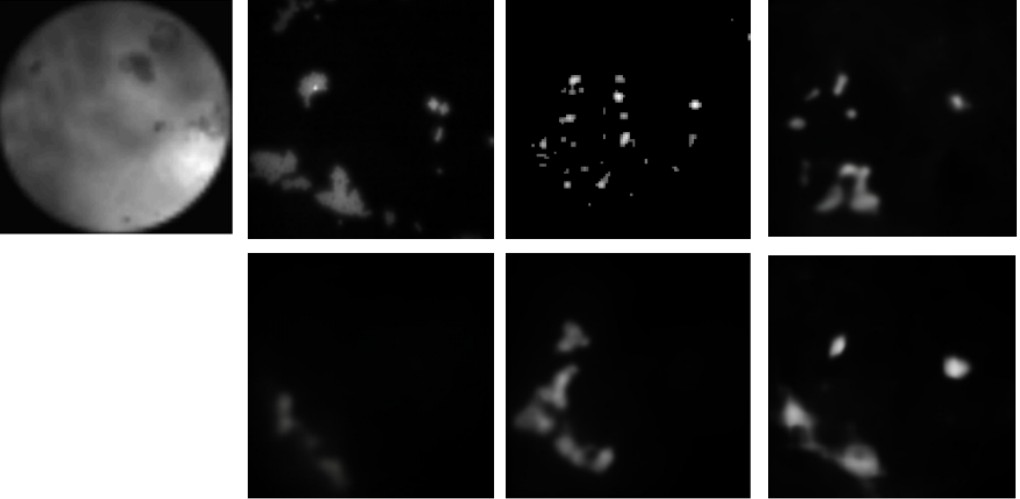

Figure 15: Example of reconstruction results with corresponding SSIM and PSNR values. Top row, from left to right: response $\mathbf{y}$, target $\mathbf{x}$, OMP $\{0.301, 68.723dB\}$, and U-Net $\{0.779, 71.691dB\}$. Bottom row, from left to right: TransUnet $\{0.672, 67.236dB\}$, Restormer $\{0.752, 71.762dB\}$, and TRUST $\{0.862, 72.744dB\}$ .

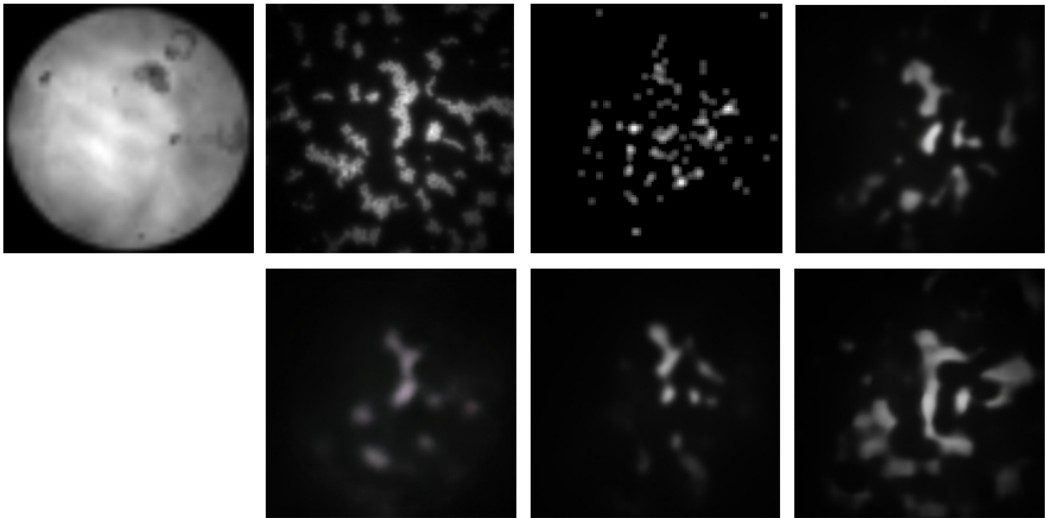

Figure 16: Example of reconstruction results with corresponding SSIM and PSNR values. Top row, from left to right: response $\mathbf{y}$, target $\mathbf{x}$, OMP $\{0.325, 63.071dB\}$, and U-Net $\{0.636, 66.712dB\}$. Bottom row, from left to right: TransUnet $\{0.553, 66.351dB\}$, Restormer $\{0.625, 66.583dB\}$, and TRUST $\{0.671, 68.276dB\}$.

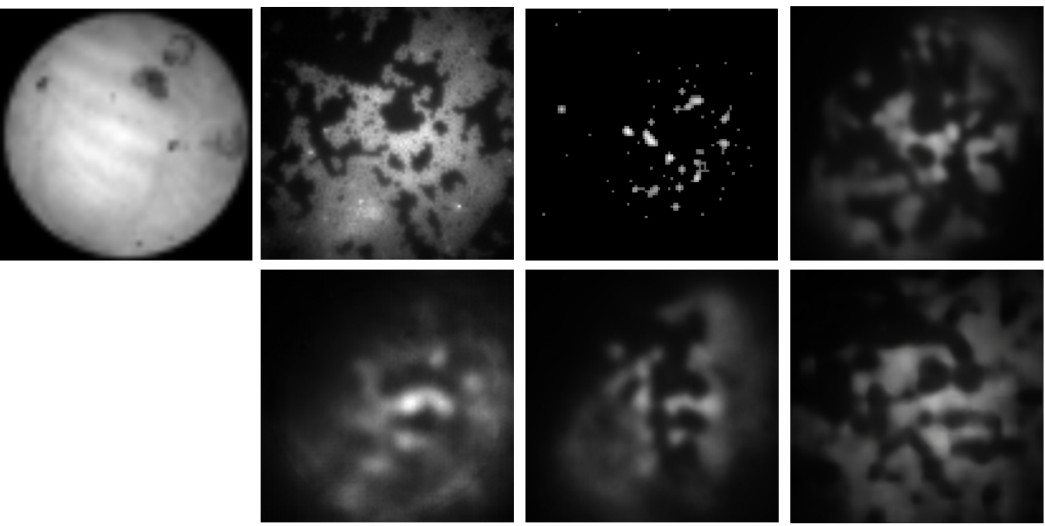

Figure 17: Example of reconstruction results with corresponding SSIM and PSNR values. Top row, from left to right: response $\mathbf{y}$, target $\mathbf{x}$, OMP $\{0.244, 58.232dB\}$, and U-Net $\{0.513, 62.105dB\}$. Bottom row, from left to right: TransUnet $\{0.409, 61.812dB\}$, Restormer $\{0.542, 62.503dB\}$, and TRUST $\{0.592, 63.427dB\}$.

### C.3 EXTENDED RESULTS ON SPARSE RECOVERY OF FASTMRI DATA

This section presents an extended comparison of sparse recovery performance on the FastMRI dataset across four deep neural network architectures.

Figures 18, 19, and 20 showcase representative examples of MRI image reconstruction under typical k-space undersampling scenarios. The corresponding quantitative results are summarized in Table 6, which reports the mean and standard deviation of recovery performance across approximately 3,000 test images.

Consistent with earlier findings, our proposed hybrid model TRUST outperforms all competing approaches in both objective and subjective measures. It achieves higher reconstruction quality as

Table 5: Average recovery performance on the optics dataset: mean $\pm$ standard deviation

| Method | MSE | MAE | PSNR (dB) | SSIM | FDR ($\times 10^{-2}$) |
|--------|-----|-----|-----------|------|------------------------|
| OMP | $0.0111 \pm 0.0032$ | $0.0435 \pm 0.0062$ | $68.04 \pm 2.03$ | $0.279 \pm 0.035$ | $5.30 \pm 1.03$ |
| U-Net | $0.00451 \pm 0.0022$ | $0.0398 \pm 0.012$ | $70.76 \pm 2.00$ | $0.772 \pm 0.053$ | $1.14 \pm 0.16$ |
| TransUNet | $0.00911 \pm 0.0040$ | $0.0440 \pm 0.012$ | $69.84 \pm 1.92$ | $0.636 \pm 0.091$ | $2.61 \pm 3.1$ |
| Restormer | $0.00823 \pm 0.0041$ | $0.0405 \pm 0.013$ | $70.48 \pm 2.13$ | $0.715 \pm 0.056$ | $0.907 \pm 0.36$ |
| **TRUST** | $\mathbf{0.00431 \pm 0.0013}$ | $\mathbf{0.0253 \pm 0.0073}$ | $\mathbf{71.992 \pm 1.94}$ | $\mathbf{0.814 \pm 0.069}$ | $\mathbf{0.901 \pm 0.22}$ |

measured by standard metrics and produces visibly more faithful image details – highlighting the effectiveness of TRUST's architecture in capturing both global structure and fine-grained spatial information in complex medical imaging tasks.

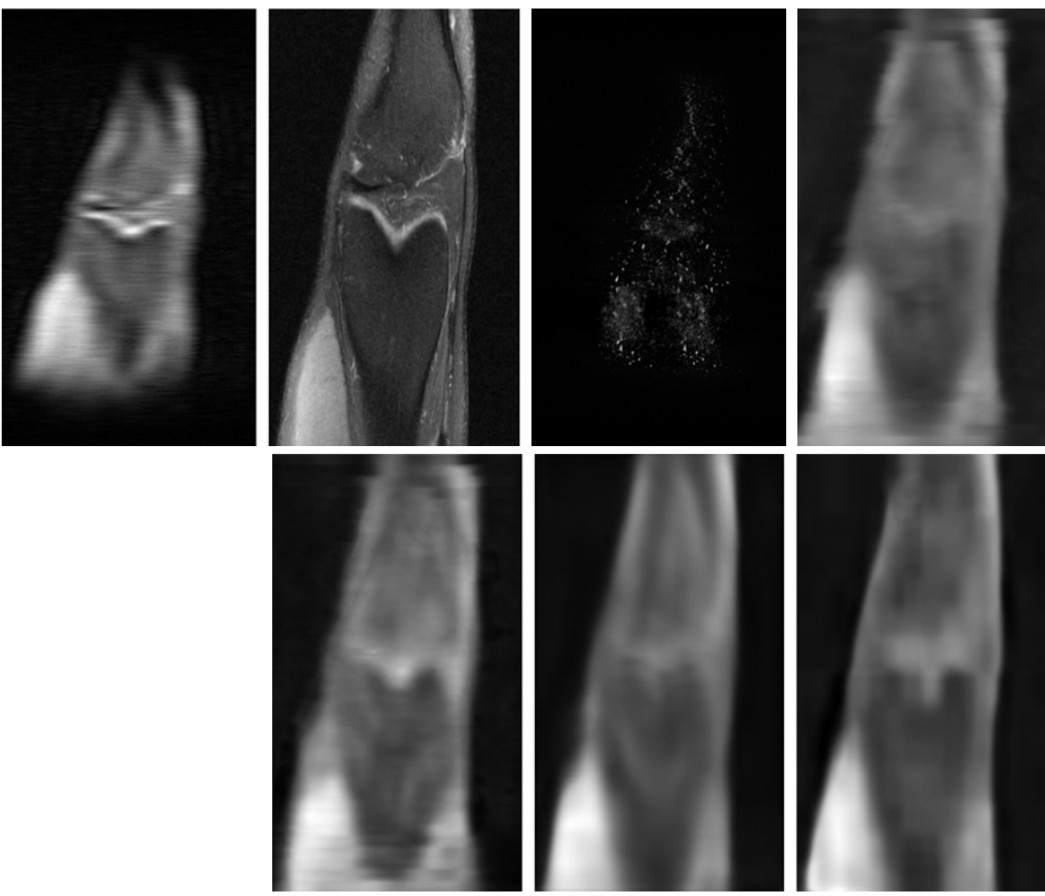

Figure 18: Example of reconstruction results with corresponding SSIM and PSNR values. Top row, from left to right: undersampled input $\mathbf{y}$, target $\mathbf{x}$, OMP $\{0.173, 15.682dB\}$, U-Net $\{0.610, 21.623dB\}$. Bottom row, from left to right: TransUnet $\{0.614, 21.956dB\}$, Restormer $\{0.623, 22.631dB\}$, and TRUST $\{0.629, 22.893dB\}$

.

# D  MODEL AND COMPUTATIONAL COMPLEXITY COMPARISON

In this section, we provide a brief supplemental comparison of the model complexity and computational efficiency of four competing deep neural network architectures: TRUST, TransUNet, Restormer, and U-Net.

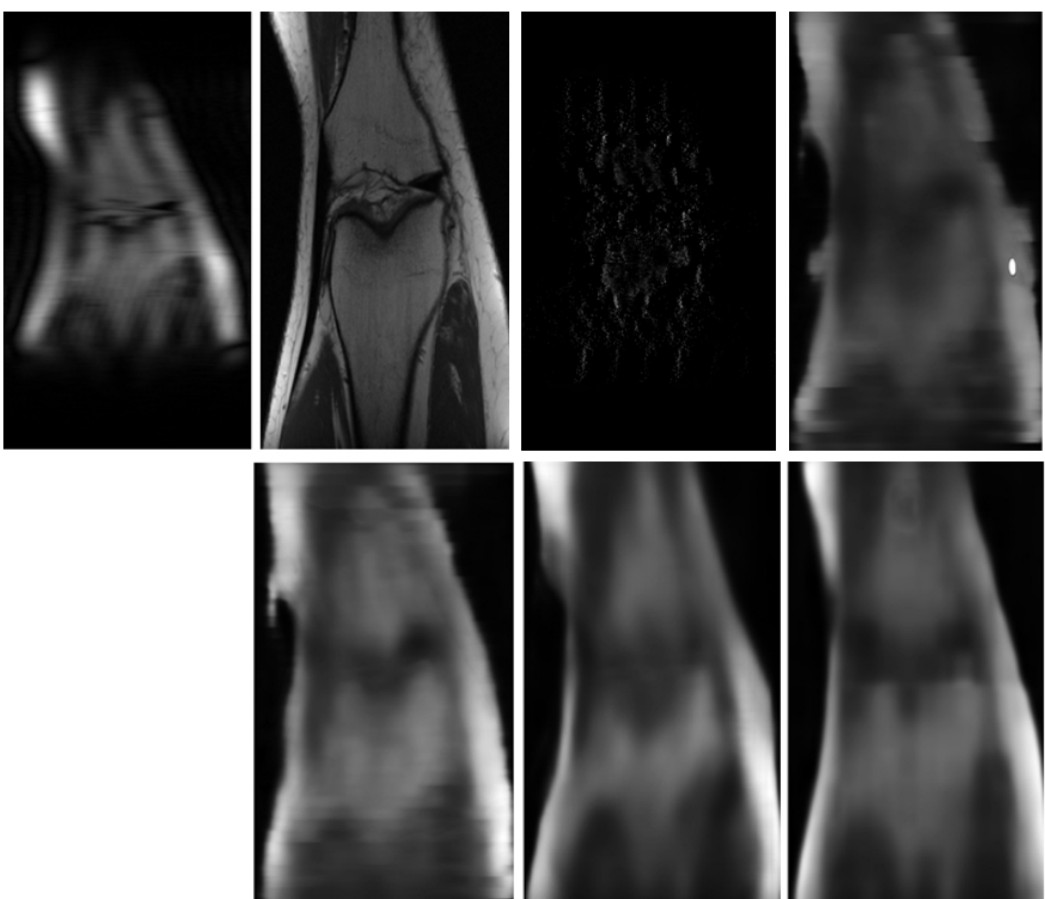

Figure 19: Example of reconstruction results with corresponding SSIM and PSNR values. Top row, from left to right: undersampled input **y**, target **x**, OMP $\{0.2430, 12.812dB\}$, U-Net $\{0.612, 18.844dB\}$. Bottom row, from left to right:: TransUnet $\{0.635, 19.593dB\}$, Restormer $\{0.636, 20.271dB\}$, and TRUST $\{0.687, 21.593dB\}$

.

Table 6: Average recovery performance on the FastMRI dataset: mean $\pm$ standard of deviation

| Method | MSE | MAE | PSNR (dB) | SSIM | FDR($\times 10^{-2}$) |
|---|---|---|---|---|---|
| **OMP** | $0.109 \pm 0.543$ | $0.138 \pm 0.0923$ | $14.37 \pm 4.34$ | $0.145 \pm 0.0395$ | $6.26 \pm 3.22$ |
| **U-Net** | $0.0861 \pm 0.0246$ | $0.0506 \pm 0.0174$ | $21.70 \pm 2.74$ | $0.668 \pm 0.0900$ | $4.26 \pm 4.99$ |
| **TransUNet** | $0.0703 \pm 0.0208$ | $0.0396 \pm 0.0178$ | $21.07 \pm 2.34$ | $0.6553 \pm 0.0863$ | $5.93 \pm 6.21$ |
| **Restormer** | $0.0692 \pm 0.0227$ | $0.0411 \pm 0.0160$ | $23.72 \pm 3.15$ | $0.698 \pm 0.0953$ | $2.97 \pm 4.74$ |
| **TRUST** | $\textbf{0.0613} \pm \textbf{0.0220}$ | $\textbf{0.0353} \pm \textbf{0.0133}$ | $\textbf{24.81} \pm \textbf{3.13}$ | $\textbf{0.717} \pm \textbf{0.0851}$ | $\textbf{2.78} \pm \textbf{4.33}$ |

While the TRUST model demonstrates strong performance across all tasks presented in previous sections, its reliance on the ViT-base backbone results in a relatively high parameter count of approximately 9 million, which is comparable to TransUNet. In contrast, Restormer maintains a smaller footprint at 3 million parameters, and U-Net remains the most lightweight, with only 2 million parameters.

In terms of training complexity, TRUST, TransUNet, and U-Net exhibit similarly efficient training behavior. Using the modest hardware configuration described earlier, each model completes 50 epochs of training in approximately 24 hours. By comparison, Restormer is significantly more computationally demanding: under the same conditions, it progresses through only 8 epochs in a 24-hour period, highlighting its heavier training requirements.

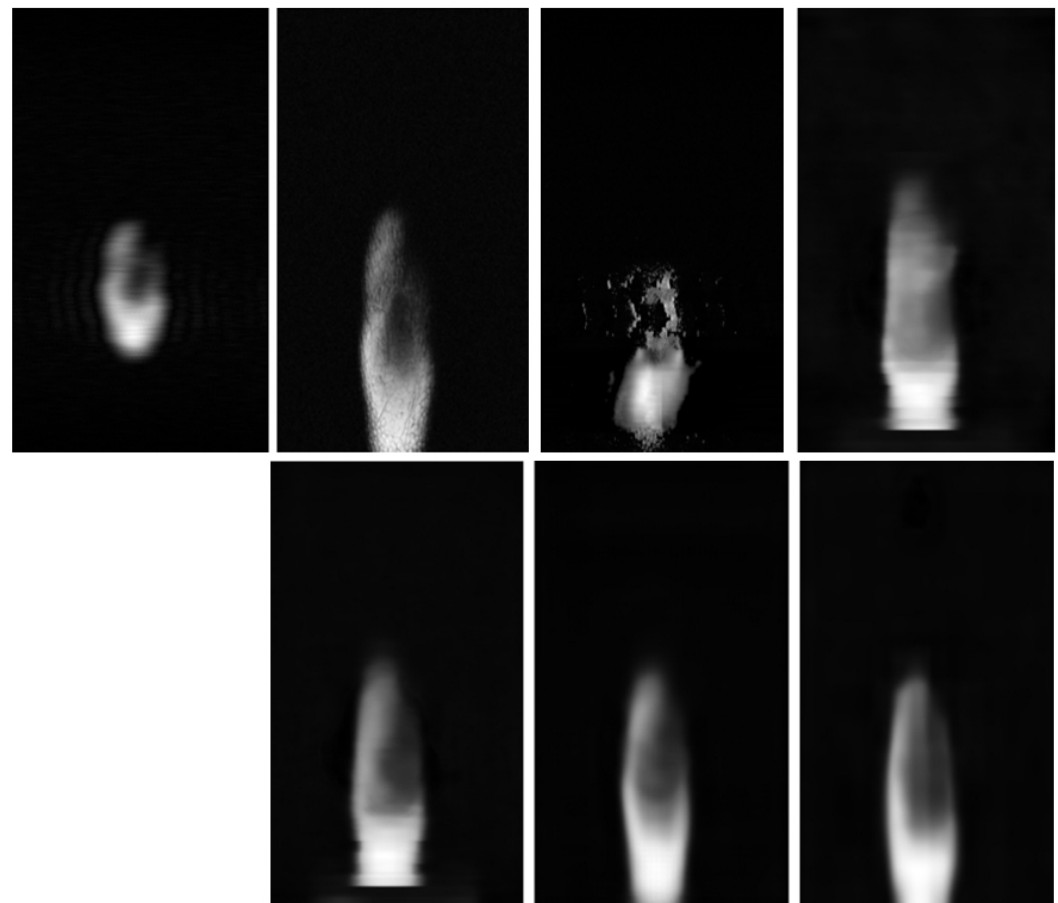

Figure 20: Example of reconstruction results with corresponding SSIM and PSNR values. Top row, from left to right: undersampled input $\mathbf{y}$, target $\mathbf{x}$, OMP $\{0.5230, 19.083dB\}$, U-Net $\{0.586, 21.693dB\}$, TransUnet $\{0.871, 22.631dB\}$, Restormer $\{0.877, 26.568dB\}$, and TRUST $\{0.889, 30.602dB\}$
.

For inference speed, U-Net is the fastest, generating images in roughly 0.006 seconds per frame, owing to its simple architecture. TRUST and TransUNet take slightly longer, averaging 0.013 seconds per image, while Restormer, with its deeper and more complex architecture, requires approximately 0.06 seconds per image.

Despite these computational trade-offs, we would like to make the following final note: the TRUST model has not yet been fully optimized. Our long-term goal is to deploy TRUST for real-time image reconstruction directly from optical system measurements. The current results suggest that reducing the computational load of the ViT-based encoder is a promising direction. In future work, we aim to explore more lightweight, task-specific attention modules that can serve as efficient substitutes for the full transformer block – potentially preserving or improving performance while significantly decreasing computational overhead.

## E    ETHICS STATEMENT.

This work adheres to the ICLR Code of Ethics. Our study uses (i) a curated subset of publicly available natural images (ImageNet) and (ii) a de-identified, publicly released MRI dataset (FastMRI) under its terms of use; no personally identifiable information is included, and no attempt at re-identification was made. We also use in-house microscopy images of fixed neuron slides that do not contain human-subject information. Consequently, this research does not involve human-subjects experiments and does not require IRB approval. We disclose potential risks: learned inverse models

may produce visually plausible but incorrect reconstructions (hallucinations) that could be harmful if used for clinical decision-making or high-stakes applications. To mitigate this, we (a) evaluate false positive structures (FDR) in addition to PSNR/SSIM, (b) report failure cases and limitations, and (c) strongly caution against deployment without domain validation and regulatory review. The datasets are used in accordance with their licenses; no sensitive attributes are inferred, and we do not train or release models to recognize protected classes. We declare no conflicts of interest and no sponsorship that could unduly influence the results.

## F REPRODUCIBILITY STATEMENT.

We took several steps to facilitate reproducibility. The paper specifies the sensing setups (fixed orthonormal transforms and subsampling masks), data preprocessing (uniform $224\times224$ resizing and patching), model architecture (ViT encoder + U-Net-like decoder), training objectives ($\ell_2$+SSIM), and evaluation metrics (PSNR, SSIM, MAE, MSE, FDR) in the main text (Section 4) and Appendix (implementation details, hyperparameters, and ablation protocols). We will provide an *anonymous* repository in the supplementary materials containing: scripts to prepare datasets, exact masks and sensing matrices used, model/config files, training and evaluation code with fixed random seeds, and commands to reproduce tables and figures. For theoretical components, all assumptions are stated and proofs are included in the Appendix. For each dataset, we reference the license/terms and describe preprocessing steps and splits. Checkpoints for the main TRUST model and baselines will be released after the review period to ensure bitwise reproducibility of reported numbers.

## G USE OF LARGE LANGUAGE MODELS

We made limited use of a large language model (LLM) strictly for *language editing* (grammar, wording, and flow) of author-written text. The LLM was not used to generate research ideas, methods, analyses, results, figures, tables, or code. All technical contributions, experimental designs, and conclusions are by the authors. Prompts contained only author-written text and non-sensitive bibliographic metadata; no private data, patient information, raw images, or dataset items were shared with the LLM. All suggestions from the LLM were reviewed and verified by the authors, and any inaccuracies were corrected. This disclosure is provided in the interest of transparency and research integrity.

