# OpenReview forum: "TRUST – Transformer-Driven U-Net for Sparse Target Recovery"
_ICLR.cc/2026/Conference — ICLR 2026 Conference Withdrawn Submission_

### Official Review · Reviewer_jJLw · 2025-10-25

**Soundness:** 1
**Presentation:** 1
**Contribution:** 1
**Rating:** 0
**Confidence:** 4

**Summary:**

The paper presents a hybrid transfomer-CNN architecture for solving inverse problems, and experiments on FastMRI, random impainting and fiber coded aperture microendoscope. The paper argues that the attention mechanism improves the capacity of the reconstruction network to deal with 'global' forward operators that mix information across large spatial regions in the image.

**Strengths:**

- the paper studies deep learning-based reconstruction methods for coded aperture microendoscopy, which seems an important application.

**Weaknesses:**

- The presentation of the paper could be significantly improved:
    - claims such as "such as classical CNNs or even U-Net—can struggle to recover globally consistent structure, especially when long-range dependencies are critical to disambiguate spatial information." are not backed by references
    - There seems to be some confusion regarding the 'attention' mechanism used in Restormer, which is based on channel attention, and not spatial self-attention used in ViT and other similar architectures.
   - various sentences are flawed, e.g. "using the same fixed mask and randomly retains 25% of pixels "
   - The setup of the optical experiments (Section 4.4) lacks more explanation about the forward operator and the imaging setting.
   - There seems to be confusion regarding unrolled networks: these architectures can use arbitrary learnable regularization terms, often being replaced by a U-Net architecture (potentially with attention blocks).

- The link with compressed-sensing, RIP, and sparsity doesn't have much relationship with the actual method that is evaluated in the experiments.
   - There is no clear "sparsity" in the proposed network, as far as I understand
   - The RIP analysis is vacuous in my opinion: self-attention is not performed directly on $y$, but rather some deep internal representation of the measurements

- The experiments seem flawed: The FastMRI results in Fig. 7 seem very blurry and severely far from those in the state-of-the-art for that level of acceleration (as it can be deduced from the measurements $y$ in that figure). It appears that the authors have also incorrectly rescaled the MRI figures, resulting in a distorted aspect ratio for the knee images.

- The novelty of the paper is limited: the paper proposes an end-to-end architecture based on existing well-known blocks.

**Questions:**

- The goal of figures 3 and 4 is unclear? What is the reader supposed to learn from them? It is not easy to take strong conclusions from feature maps and their downsampled counterparts.

- What is the goal of the CS equations in (1)? They have no link to the proposed method as far as I understand.

---

### Official Review · Reviewer_AAAK · 2025-10-31

**Soundness:** 2
**Presentation:** 3
**Contribution:** 2
**Rating:** 2
**Confidence:** 4

**Summary:**

This work proposes a hybrid architecture (TRUST) that combines a Vision Transformer (ViT) encoder with a U-Net–style decoder to address sparse signal recovery in inverse problems where the forward operator A is unknown and must be implicitly learned from observation–target pairs. The proposed architecture leverages multi-resolution attention to estimate sparse support directly from the measurements $y$, then uses attention-guided skip connections to steer the decoder toward support-consistent regions. The authors provide a theoretical justification based on the Restricted Isometry Property (RIP), showing that attention maps computed on $y$ approximate those on the target signal $x$ with an error bounded by the RIP constant.

**Strengths:**

* The authors provide a theoretical justification using the Restricted Isometry Property and show that self-attention computed on the measurements $y$ approximates that on the target signal  $x$, with error bounded by the RIP constant. This offers a signal-processing justification for applying self-attention directly in the measurement domain, a useful insight for the inverse problems community.

**Weaknesses:**

* While the hybrid encoder-decoder architecture of TRUST is well motivated, it closely resembles TransUNet. The claimed innovation, the attention-guided skip connections, is not described in sufficient technical detail (e.g., how attention maps modulate skip features), making it hard to assess how this differentiates from prior hybrid models. More specifically, the design presented in Section 3.2 and Figure 2 is not different from the standard ViT + UNeT architecture.
* The authors assume sparsity but do not clarify whether it refers to pixel-domain sparsity, transform-domain sparsity, or learned sparsity. This matters because the theoretical RIP argument applies strictly to k-sparse vectors in the same domain as A, which may not hold in natural images.
* The bound $|x^TA^TAx' - x^Tx'|$ assumes $x$ and $x^′$ are k-sparse and normalized, which is unrealistic for natural images (e.g., ImageNet). The empirical success may rely more on pretraining and data-driven learning than the RIP-based guarantee, which weakens the theoretical contribution’s practical scope.
* Models like SwinIR, Uformer, or MIMOUNet, which also integrate multi-scale attention for inverse problems are not discussed, raising questions about the completeness of the related work.
* By inspecting the code, the comparison between TRUST and TransUNet is not fair.  The encoder used in TransUNet is trained from scratch while TRUST utilizes pretrained CLIP weights.

**Questions:**

The authors repeatedly emphasize “attention-guided skip connections” as a core innovation, but never specify how attention maps modulate the skip pathways. Can you please elaborate on this aspect?

---

### Official Review · Reviewer_ejiZ · 2025-11-03

**Soundness:** 2
**Presentation:** 2
**Contribution:** 2
**Rating:** 2
**Confidence:** 4

**Summary:**

The paper proposes TRUST, a ViT-encoder + U-Net-decoder architecture for solving linear inverse problems when the forward operator A is unknown or poorly characterized. The core idea is that self-attention on measurements y=Ax can approximate attention on x under RIP-like assumptions, allowing a Vision Transformer to extract global dependencies directly from measurements. Attention-guided skip connections then feed this global context into a U-Net decoder to reconstruct fine details. Experiments on masked ImageNet patches, single-coil FastMRI, and a coded-aperture optics dataset show moderate improvements over U-Net and TransUNet, but only small or mixed gains over Restormer.

**Strengths:**

Clean, practical hybrid architecture combining ViT for global context with a U-Net decoder for detail refinement.

Pretraining benefits are well demonstrated, and the overall design is computationally efficient at inference relative to Restormer.

The use of the false discovery rate (FDR) metric is a valuable addition for quantifying hallucinations in reconstructions.

Implementation simplicity: TRUST can be applied with minimal modification to standard reconstruction pipelines.

**Weaknesses:**

heoretical contribution is minimal. The RIP-based argument about attention similarity is trivial and disconnected from end-to-end reconstruction performance.

Claimed robustness to unknown or uncalibrated A is not demonstrated; all experiments use known and fixed operators.

Improvements over Restormer are marginal or inconsistent (e.g., identical PSNR/SSIM in ImageNet 100%, mixed results at 25%, small +1 dB gain on FastMRI). No confidence intervals or statistical tests are provided.

The analysis does not identify which component—ViT encoder, skip design, or loss formulation—drives improvements.

Benchmarks are limited: masked ImageNet and single-coil FastMRI do not test generalization or operator mismatch.

No study of computational budget, model robustness, or sensitivity to operator perturbations.

**Questions:**

Can you connect the RIP-based attention bound to actual reconstruction accuracy or sample complexity?

How robust is TRUST to operator mismatch—e.g., unseen masks, changed optics PSFs, or altered sampling patterns?

Are the improvements over Restormer statistically significant under equal compute and training schedules?

Which architectural element (ViT encoder, skip connections, or loss) is primarily responsible for the gains?

Can the model generalize to multi-coil or nonlinear operators?

How reliable is the FDR metric as a measure of hallucination—does it correlate with perceptual error or SSIM/PSNR?

---

### Official Review · Reviewer_EBKL · 2025-11-07

**Soundness:** 2
**Presentation:** 2
**Contribution:** 2
**Rating:** 2
**Confidence:** 3

**Summary:**

This paper introduces a hybrid deep network, TRUST, for solving linear inverse problems where the sensing operator
𝐴 is unknown or partially known. The method uses: a Vision Transformer encoder to estimate global sparse support directly from the measurement y, and attention-guided skip connections and a U-Net-style decoder to refine local details. The method is supervised-based.
Experiments on optical coded-aperture imaging, FastMRI, and masked ImageNet show improved PSNR/SSIM and fewer hallucination artifacts compared with U-Net, TransUNet, and Restormer baselines.

**Strengths:**

+ Clear motivation, and it is important to tackle the practical challenge of incomplete knowledge of the forward operator, moving beyond the idealized assumptions of traditional compressed sensing.

+ The paper includes ablation studies on loss functions, skip connections, and ViT pretraining, providing certain insights into the contribution of each component to the studied question.

**Weaknesses:**

- The core motivation of this paper is actually very straightforward — essentially, the authors aim to reconstruct images from incomplete measurements when the sensing operator A is unknown/partial. However, the presentation makes this simple idea unnecessarily difficult to follow. The writing is overly verbose and circuitous: the abstract and introduction repeat the same points in different forms instead of directly stating the motivation and contribution. As a result, readers must work hard to extract a concept that could have been summarized in a few sentences. Also, for example, Figure 1 and its accompanying discussion add little technical value and could be safely removed; the space would be better used to clearly explain the problem setting and the novelty, and include more evaluation tasks.
- Technically, the work has limited novelty. The model is essentially a supervised combination of a Vision Transformer encoder and a U-Net decoder, using standard losses (ℓ₂ + SSIM) and skip connections. The method remains a straightforward supervised mapping. Conceptually, this is an incremental variation of prior architectures such as TransUNet or Restormer.
- Generalization and reconstruction quality remain weak. Results show visible blur and artifacts, particularly outside the training domain, indicating that the network overfits to specific measurement operators and does not robustly handle unseen conditions.
- The authors should also consider evaluating Helmholtz-governed wave problems. This is a natural fit for their “unknown A" claim and would more rigorously probe operator-shift generalization, complex-valued reconstruction, and artifact control.

**Questions:**

See "weaknesses"

---

### Note · Authors · 2025-11-15

I have read and agree with the venue's withdrawal policy on behalf of myself and my co-authors.